# Preparation of Nanocellulose Whisker/Polyacrylamide/Xanthan Gum Double Network Conductive Hydrogels

**Zhiwei Du, Yalei Wang and Xiurong Li \***

College of Resources, Environment and Materials, Guangxi University, Nanning 530004, China
\* Correspondence: rongrong1994@gxu.edu.cn

**Abstract:** Hydrogels' poor mechanical and recovery characteristics inhibited their application as a plastic deformable three-dimensional cross-linked network polymer with electrical properties for intelligent sensing and human motion detection. Cellulose has also been added to the hydrogel to enhance its mechanical properties. The hydrogel has been enhanced this way, and the double-network hydrogel has superior recovery and mechanical capabilities. This study used the traditional free radical polymerization method to prepare double-mesh hydrogels, with polyacrylamide as the backbone network, xanthan gum double-helix structure, and $Al^{3+}$ complex structure as the second cross-linked network, and endowing the hydrogels with good mechanical recovery and mechanical properties. Adding cellulose nanowafers (CNWs) improved the mechanical properties of the hydrogels. The hydrogel could detect body movements and various postures in the same environment. Moreover, the hydrogel has excellent recovery, mechanical properties, and tensile strain; the maximum fracture stress is 0.14 MPa, and the maximum strain is 707.1%. In addition, Fourier infrared spectroscopy (FTIR) of xanthan gum and Xanthan gum—$Al^{3+}$ were analyzed, and thermogravimetric analysis (TGA) and LCR bridge were used to analyze the properties of hydrogels. Notably, hydrogel-based wearable sensors have been successfully constructed to detect human movement. Its mechanical properties, sensitivity, and wide range of properties make hydrogel a great potential for various applications in wearable sensors.

**Keywords:** double network hydrogels; xanthan gum; polyacrylamide; cellulose nanowhiskers; strain sensors; recovery properties

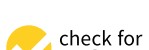



## 1. Introduction

The preferred material for tissue scaffolds is hydrogel, a polymer with a three-dimensional cross-linked network structure and much water. Hydrogels are similar in structure and chemistry to soft tissues, including cartilage and tendons [1,2]. Likewise, conductive hydrogels are created by mixing them with conductive materials, such as polyaniline [3], polypyrrole [4], poly(3,4-ethylenedioxythiophene)/polystyrene sulfonic acid (PEDOT/PSS) [5], Mexen [6], carbon nanotubes [7], and graphene oxide [8], giving cellulose good conductive properties. Hydrogels have, consequently, attracted much interest in smart sensing [9,10], wearable devices [11–13], and tissue engineering [14,15]. Resistive [16] and capacitive [17] applications of conductive hydrogels can be further separated. To perceive the movement of an object and determine the motion of individual components of the object by the difference in resistivity ($R/R_0 = (R - R_0)/R_0$), resistive sensors primarily use the deformation of hydrogels (stretching and compression) [18–20]. On the other hand, capacitive sensors sandwich a hydrogel between two electrodes and utilize deformation to change the hydrogel's capacitance to detect and identify an object's motion based on the differences in the rate at which the capacitance changes ($C/C_0 = (C - C_0)/C_0$) [21,22]. Polymers that make up the network's backbone, such as polyacrylamide (PAAM) [23], polyvinyl alcohol (PVA) [24], polyacrylic acid (PAA) [25], and others, make up most hydrogels worldwide. However, these hydrogels still require hard work on their mechanical and recovery properties. A second cross-linked network was added to the hydrogels to enhance the mechanical

and recovery capabilities, creating "double-network" (DN) hydrogels. Among the many examples are P(AA-co-AAm)/PVA DN hydrogels [26], Mexen composite PVA/CMC DN hydrogels [27], PVA/PAAM DN hydrogels [28], PAAM/CA DN hydrogels [29], and DN hydrogels with gelatin and Na+ electrostatic networks [30]. These hydrogels are now more widely employed since their mechanical and recovery qualities have improved to a certain extent. Cellulose is one of the most common organic polymers on earth, with high yield, renewable and non-toxic, non-hazardous, and biodegradable properties, making it widely used. Three types of nanocellulose have been proposed from cellulose: cellulose nanocrystals (CNCs), cellulose nanofibers (CNFs) and bacterial cellulose (BNC). The former is prepared by strong acid hydrolysis and ionic liquid dissolution of CNCs, while the latter is prepared by the high-pressure homogenization method, microsulfuration, and nano-grinding of CNFs [31]. CNFs are flexible, while CNCs improve the hydrogel's mechanical properties [32]. Pi et al., prepared CNC/Mxene PVA/PAAM hydrogels with desirable strain-sensitive conductivity and good thermal resistivity [33]. Cheng et al. used microwave incorporation to prepare PAA/CNC hydrogels with good biocompatibility and high transmission to ultrasound [34]. Peng et al., cross-linked hydrogels with a high swelling rate using quaternized membranous cellulose nanocrystals (Q-TCNCs), carboxymethylcellulose (CMC), and epichlorohydrin (EPI), and their mechanical properties and swelling rate could be controlled by adjusting the Q-TCNCs content [35]. Lai et al., prepared cross-linked network hydrogels of CNCs or cationic cellulose nanocrystals (CCNCs)/AA/DMAPS polymerized with $Al^{3+}$, where the modified cellulose imparted high toughness while enabling them to acquire 3D printing properties [36]. CNCs improve the mechanical properties of hydrogels because they are easily modified with other functional groups due to their many hydrophilic and hydroxyl groups (-OH), giving them a variety of properties. Adding cellulose to the hydrogel can improve the hydrogel's water absorption and swelling rate. CNC-dopamine (DOPA) and PAAM hydrogels were prepared according to Howson et al.; adding cellulose improved the hydrogel's mechanical properties and the water absorption and swelling rate [37]. Xanthan gum has a double helix structure that can improve the mechanical structure of the hydrogel, and xanthan gum can be complex with $Fe^{3+}$ under weak acid conditions to form a cross-linked network [38].

This paper uses PAAM as the first cross-linked network and backbone network, while xanthan gum forms a second cross-linking network by complexing its carboxyl group with $Al^{3+}$. Since xanthan gum has a double helix structure and can be complexed with $Al^{3+}$ with a carboxyl group, it can be used as a second network to enhance the mechanical properties of hydrogels. The mechanical properties of hydrogels were further improved by adding crystalline cellulose nanocrystalline whiskers (CNWs). PAAM/Xanthan gum— $Al^{3+}$/CNWs DN hydrogel was formed. The DN hydrogel studied in this paper has good recovery efficiency, good fracture stress (up to 0.14 MPa), good fracture stress (up to 707.1%), and good electrical conductivity. Hydrogel has a wide range of applications because of its excellent mechanical properties, good recovery performance, and fast reaction.

## 2. Experimental Section

### 2.1. Experimental Materials

Acrylamide (AAm, MW 71.08, analytical purity 99.0%) and anhydrous aluminum chloride ($AlCl_3$, analytical purity 99%) were purchased from Shanghai Maclean Biochemical Technology Co. (Shanghai, China) Sodium persulfate ($Na_2S_2O_8$, analytical purity), N,N′-methylenebisacrylamide (MBA, analytical purity), and sodium chloride (NaCl, analytical purity) were purchased from Tianjin Damao Chemical Reagent Factory (Tianjin, China). Cellulose nanowhiskers (CNWs, 20–30 nm in diameter, 500–600 nm long, a crystallinity of 85%, liquid, and 5% solid content (5 wt.%)) were purchased from Northern Century (Jiangsu) Cellulose Materials Co (Nanjing, China). Xanthan gum (GX, USP grade) was purchased from Aladdin (Shanghai, China).

## 2.2. Preparation of PEC Hydrogels

0.2 g of NaCl and 0.08 g of GX were dissolved in 30 (28, 26, 24) mL of deionized water, and mixed by stirring for 2 h. 2 mL AlCl$_3$ (1 wt.%) was added dropwise under vigorous stirring. After mixing, 8 g of acrylamide (AAm), 80 mg of sodium persulfate (SPS), 4 mg of N,N'-methylenebisacrylamide (MBA), and nanocellulose fibers (5 wt.%) (CNWs) were added. After mixing, the mixture was injected into a polytetrafluoroethylene (PTFE) mold weight and reacted in an oven at 50 °C for 2 h to obtain a nanocellulose double network conductive hydrogel (the hydrogel was 5 mm high, 15 mm wide and 40 mm long). Definition of hydrogel PEC$_x$: P stands for polyacrylamide, E stands for GX, C stands for cellulose nanofibers, x is the mass ratio of added nanocellulose to acrylamide, so x value is 1.25%, 2.5%, and 3.75%. The specific dosage of cellulose nanofiber double network conductive hydrogels is shown in Table 1.

**Table 1.** Components of cellulose nanofiber double network conductive hydrogels.

| Code | NaCl (g) | GX (g) | AlCl$_3$ (1 wt%) (mL) | CNWs (5 wt%) (g) | AAm (g) | SPS (mg) | MBA (mg) | Water (mL) |
|---|---|---|---|---|---|---|---|---|
| P | 0 | 0 | 0 | 0 | 8 | 80 | 4 | 32 |
| PE | 0.2 | 0.08 | 2 | 0 | 8 | 80 | 4 | 30 |
| PEC$_{1.25\%}$ | 0.2 | 0.08 | 2 | 2 | 8 | 80 | 4 | 28 |
| PEC$_{2.5\%}$ | 0.2 | 0.08 | 2 | 4 | 8 | 80 | 4 | 26 |
| PEC$_{3.75\%}$ | 0.2 | 0.08 | 2 | 6 | 8 | 80 | 4 | 24 |

## 2.3. Sensing Properties of PEC Composite Conductive Hydrogels

The change in resistance of the PEC composite conductive hydrogel was measured using an LCR bridge tester. The rate of change in resistance ($\Delta R/R_0$) of the PEC composite conductive hydrogel was defined as:

$$\frac{\Delta R}{R_0} = \frac{R - R_0}{R_0} \tag{1}$$

where $R_0$ and $R$ represent the initial resistance of the PEC composite conductive hydrogel and the resistance after deformation, respectively.

## 2.4. Swelling Properties of PEC Composite Conductive Hydrogels

The swelling properties of the hydrogels were measured by weighing them at room temperature; a certain amount of the hydrogel was weighed and soaked in beakers filled with water until swelling equilibrium was reached. The masses of the hydrogels before and after swelling were weighed on an electronic balance. The swelling rate (SR) of the hydrogel is calculated using the following equation:

$$SR = \frac{W_s - W_d}{W_d} \tag{2}$$

where $W_s$ and $W_d$ represent the masses before equilibrium and swelling, respectively.

## 2.5. Instrumentation and Characterization

GX solution and GX-AL$^{3+}$ mixed solution were detected by a Fourier infrared spectrometer. The measuring wavelength range was 400–4000 cm$^{-1}$. A new high-resolution field emission scanning electron microscope (SEM) was used to observe the microstructure of 10 kV. The lyophilized hydrogels: the hydrogel samples were freeze-dried for 24 h and then gold-sprayed to observe the microstructure at 10 kV. The Al$^{3+}$ distribution was analyzed using an energy spectrum analyzer (EDS). Thermogravimetric analysis (TGA) (TG–DSC) was used to analyze PEC composite hydrogels and their thermal stability differ-

ences in the range of 20–790 °C were compared. The mechanical properties of the hydrogels were examined using a small mechanical testing machine (tensile rate: 10 mm/min)

## 3. Results and Discussion

### 3.1. Formation of PEC Hydrogels

This study uses in situ free radical polymerization to create PEC composite conductive hydrogels in a straightforward one-pot approach. Figure 1a shows that adding NaCl and AlCl$_3$ to GX makes the carboxyl group on GX undergo a complex reaction with Al$^{3+}$. Figure 1b is the FTIR spectrum of GX and GX-Al$^{3+}$. The characteristic peaks of GX at 3408 and 2926 cm$^{-1}$ are O–H stretching vibration and C-H stretching vibration, respectively. At 1727 and 1614 cm$^{-1}$, the stretching vibration of the carbonyl group (C=O) representing the acetyl group and the asymmetric stretching vibration of the carboxyl group (C=O) appears at GX-Al$^{3+}$ at 1636 cm$^{-1}$, which proves the coordination between carboxyl acid and Al$^{3+}$. Figure 2 shows that cellulose nanowhisker dispersion, acrylamide, and ammonium persulfate initiator (N,N′-methylene diacrylamide) were added into the GX-Al$^{3+}$ solution and fully stirred. It was then transferred to a mold and used for heat-induced in situ radical polymerization to generate PEC composite conductive hydrogels. PEC composite conductive hydrogels comprise the cross-linking network, metal coordination bonds, and dynamic hydrogen bonds. The first cross-linking network is the cross-linking network of polyacrylamide, and the second cross-linking network is the complex bond between Al$^{3+}$ and the carboxyl group in GX. There are hydrogen bonds between CNWs, acrylamide, and GX. Because GX has a helical structure, which enhances hydrogels' mechanical and recovery properties, its carboxyl group physics causes a complex network with Al$^{3+}$. Al$^{3+}$ is a physical cross-linking agent in a PEC composite conductive hydrogel system. In the hydrogel system, Na$^{+}$ and Cl$^{-}$ provide sufficient ion migration, enabling the composite hydrogel to conduct electricity.

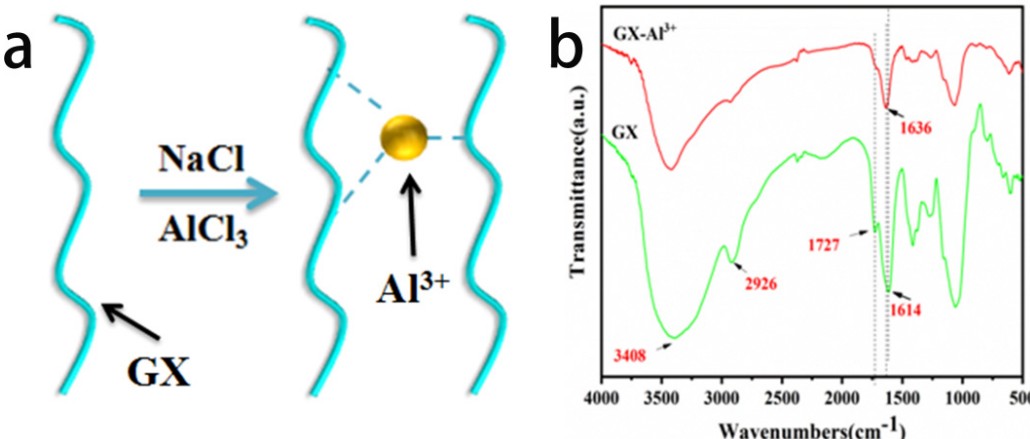

**Figure 1.** (**a**) Schematic diagram of GX-Al$^{3+}$ cross-linking and (**b**) FTIR spectra of GX and GX-Al$^{3+}$.

### 3.2. Microstructure of PEC Composite Conductive Hydrogels

SEM was used to analyze the microscopic surface morphology of PEC composite conductive hydrogel after lyophilization. For hydrogel morphology analysis, the hydrogel is nucleated by freezing in the first freeze-drying step, and the solute is concentrated between crystal growth. The second step directly sublimates the ice at low pressure, avoiding the intermediate liquid phase and leaving a porous structure often used to make porous polymer materials [39]. As shown in Figure 3a, the microstructure of PAAM polymer hydrogels is a smooth surface with a porous structure formed by the growth direction of the secondary crystals perpendicular to the primary crystals. As shown in Figure 3b, the pore size increases after adding GX, possibly due to the water absorption of GX, increasing the surrounding water, which leads to crystal growth and, thus, pore size increase. As shown in Figure 3c, the pores become denser after adding

cellulose. Figure 3d shows that when 6 g of cellulose nanowhiskers were added, cellulose cross-linked entanglement occurred due to the high cellulose concentration. As shown in Figure 2e,f, in PEC composite conductive hydrogel, $Al^{3+}$ is uniformly distributed in the gel skeleton network, Figure 1, indicating that $Al^{3+}$ performs a uniform complexation reaction with carboxyl groups in GX.

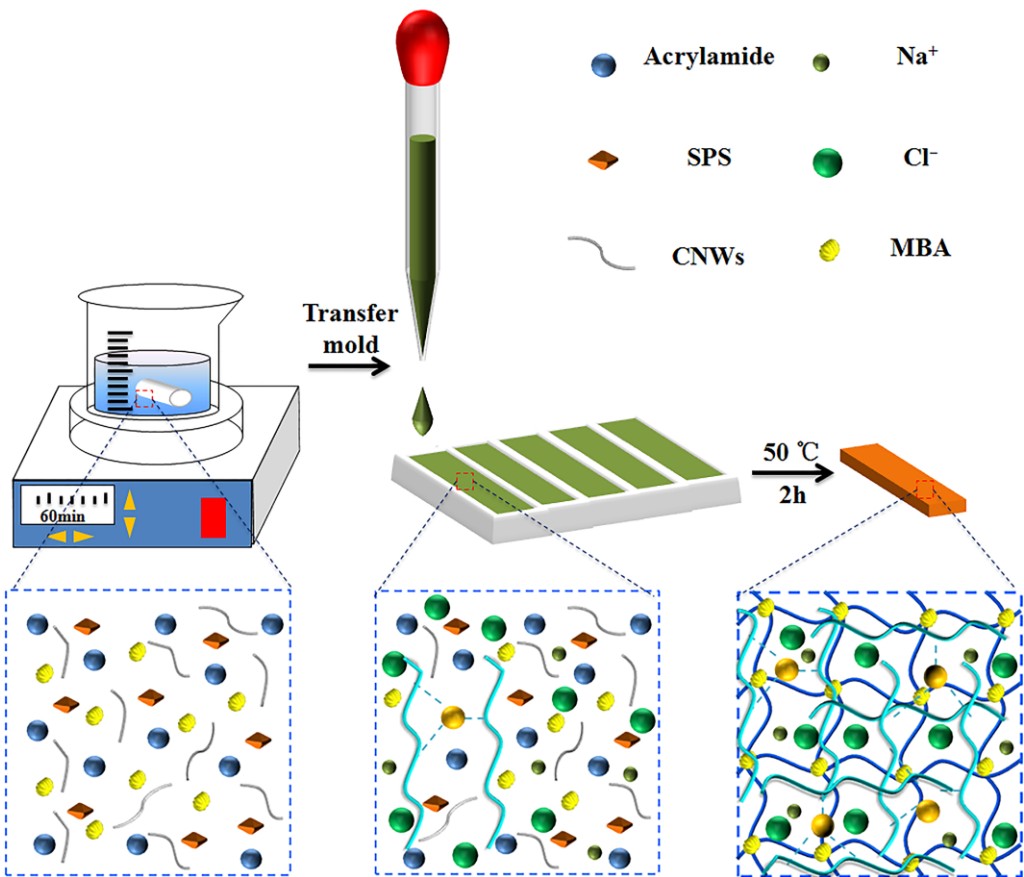

**Figure 2.** Flow chart of PEC hydrogel formation.

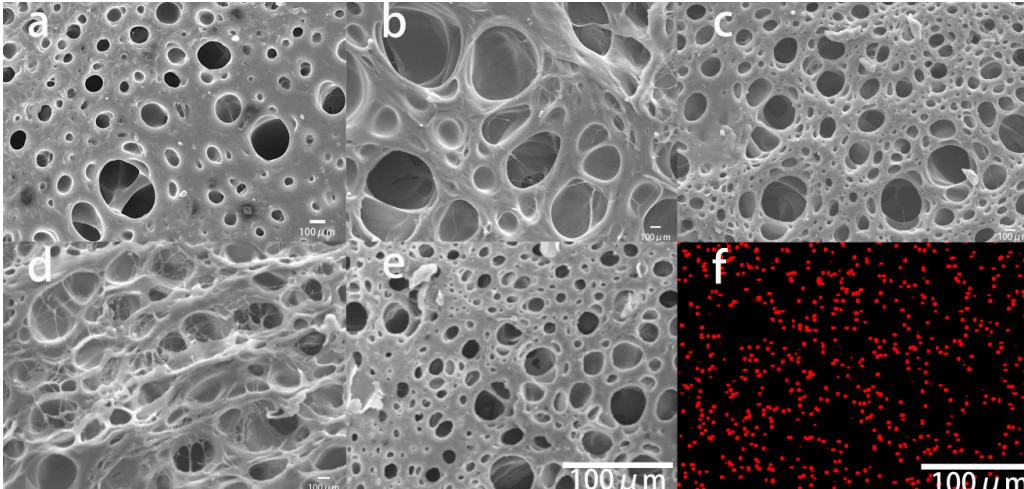

**Figure 3.** SEM images of PEC composite hydrogels after freeze-drying (**a**) P, (**b**) PE, (**c**) PEC$_{2.5\%}$, (**d**) PEC$_{3.75\%}$, (**e**) SEM images taken by EDS, (**f**) EDS images of $Al^{3+}$.

### 3.3. TGA of PEC Composite Hydrogels

In order to further study the influence of GX and CNWs on hydrogels, the thermal stability of P, PE and PEC$_{3.75\%}$ hydrogels was studied, as shown in Figure 4. At the weight loss range of 20–180 °C, due to the presence of a large number of free water molecules in the hydrogel sample, the water evaporates and the mass of each component decreases rapidly. In the weight loss range of 180–350 °C, the mass of each component decreases slowly, which is caused by the volatilization of the bound water in the hydrogel sample. At the weight loss range of 350–450 °C, the degradation of polymer segments and carbonization of carbon-containing functional groups in hydrogel causes the reduction of the mass of each component. The low mass loss of PEC$_{3.75\%}$ hydrogel is most likely due to the increase of chemical bonds in the hydrogel network and the increase of hydrogen bond content due to the introduction of CNWs.

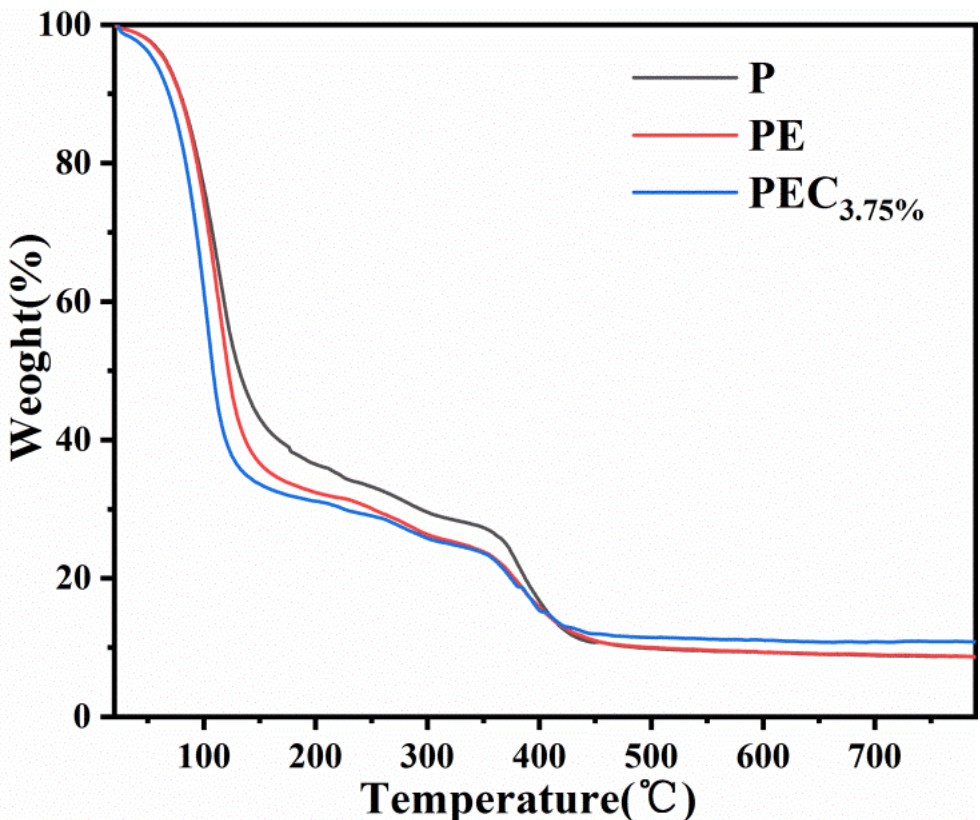

**Figure 4.** TGA spectra of P, PE and PEC$_{3.75\%}$ (temperature programmed, 20–790 °C).

### 3.4. Mechanical and Swelling Properties of PEC Composite Hydrogels

PEC composite conductive hydrogels have excellent mechanical properties and can withstand various deformations such as stretching, twisting, and knotting (Figure 5a–c), up to four times their original length. This may be because the PEC composite conductive hydrogel is a double-network conductive adhesive, where the easily ductile network is physically cross-linked, which helps the double-network conductive hydrogel retain its tensile properties.

To demonstrate the role of CNWs on the dynamic properties of energy dissipation within the hydrogel, a series of uniaxial tensile tests at a tensile speed of 10 mm/min were conducted to display the mechanical properties of PEC composite conductive hydrogels, which are critical for their applications. The effect of different CNWs concentrations on the mechanical properties of PEC composite conductive hydrogels was investigated. As shown in Figure 6a, the tensile stress of the hydrogel formed by AAM polymerization alone was 0.0846 MPa, and the elongation at break was 475%. When GX was added, the tensile stress and elongation at break were 0.912 MPa and 615.4%, respectively. GX

significantly increased tensile stress and elongation at break, possibly because GX has a three-dimensional helical structure and the carboxyl group and $Al^{3+}$ form a complex structure. The force between the carboxyl group and the coordination bond formed by $Al^{3+}$ is small, so when CNWs were added, the tensile stress increased, and the elongation at break increased, then decreased with the increase in the concentration of CNWs. When 6 g of CNWs was added, the tensile stress of the PEC composite conductive hydrogel reached 0.1409 MPa, 1.67 and 1.55 times higher than that of the PAAM hydrogel and the PAAM hydrogel with GX, respectively. The cross-link density increased with the concentration of CNWs, and the increase in cross-link density had a positive and then a negative effect on the toughness of the PEC composite conductive hydrogel (Figure 3d). The maximum elongation at a break of 707.1% was achieved at 4 g of CNWs, 1.49 and 1.15 times higher than the previous two, respectively. With the further addition of CNWs, the elongation at break of the PEC composite conductive hydrogel decreased to 656.6%. The reason for the decrease of elongation at break is that when the concentration of CNWs reaches a certain level, the cross-linking density in the hydrogel is too high.

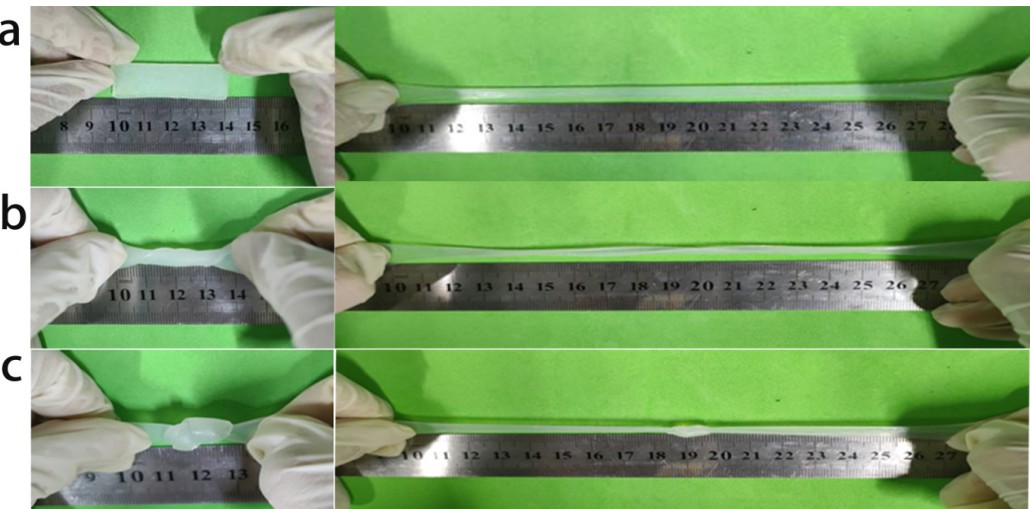

**Figure 5.** Composite hydrogels exhibit excellent deformation under (**a**) stretching, (**b**) twisting, and (**c**) knotting conditions.

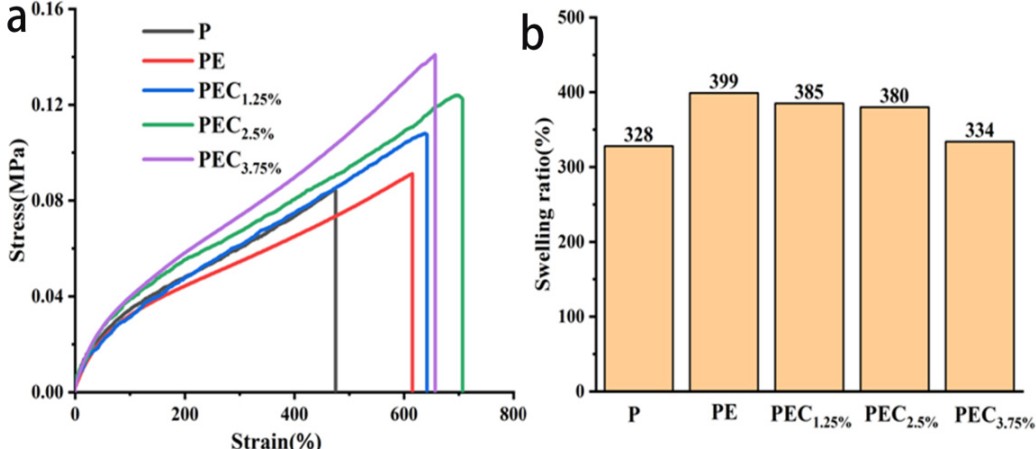

**Figure 6.** (**a**) Tensile stress-strain curves of P, PE, and PEC hydrogels (**b**) Water absorption and swelling rates of P, PE, and PEC hydrogels.

The cross-linking density inside the hydrogel network determined the structural characteristics and swelling properties of the PEC composite conductive hydrogels. The larger the cross-linking density inside the hydrogel network, the smaller the swelling degree. As shown in Figure 6b, when GX is added, the hydrogel's swelling rate increases due to the good water absorption performance of GX. However, with the addition of CNWs, the swelling rate of the PEC composite conductive hydrogel decreased, mainly because CNWs formed a closer network structure with GX and PAAM through hydrogen bonding, decreasing the swelling degree. Meanwhile, the swelling degree of the PEC composite conductive hydrogel decreased from 380% to 334% when the CNWs increased from 4 to 6 g. This indicates that $PEC_{3.75\%}$ composite conductive hydrogels have tighter network structures than other hydrogels.

### 3.5. Conductive Properties of PEC Composite Conductive Hydrogels

To test the sensing performance of PEC composite conductive hydrogels, this paper uses an electrical bridge to measure their electrical conductivity. The NaCl and $AlCl_3$ in the PEC composite conductive hydrogel release many free ions; therefore, the hydrogel has good electrical conductivity. As shown in Figure 7a, the hydrogel maintained good stability after five stretching cycles at specific stretching multipliers (20%, 40%, and 80%), indicating that the hydrogel maintains stable conductive properties over a certain period. As shown in Figure 7b,c, the conductive chemical properties of the hydrogel also remain stable when stretched to a certain multiplicity. The conductive properties of the hydrogel increase smoothly under increasing circumstances, indicating that the hydrogel can maintain stable conductive properties at relatively high (greater than 100%) and low (less than 100%) stretching multiplicities. When the stretching reaches four times its size, the rate of change of resistance ($\Delta R/R$) continues to rise steadily. When reached, it remains stable. As shown in Figure 7d, the response time of the hydrogel at high stretching multiples is 591 ms (response time refers to the time required for a force to be applied to the hydrogel until the hydrogel receives the reaction result), indicating that the hydrogel has a fast response rate. The rapid response of the hydrogel is mainly reflected in the rapid movement of free ions in the hydrogel's network. As shown in Figure 7e, the power supply, wire, PEC composite conductive hydrogel, and LED bulb are connected in series to form a closed circuit, and the LED bulb is illuminated. As shown in Figure 7f, the PEC composite conductive hydrogel was stretched. At the same time, the LED bulb was still lit, proving that the PEC composite conductive hydrogel has excellent electrical conductivity. In contrast, the brightness of the LED bulb decreased, indicating that the resistance of the PEC composite conductive hydrogel was increasing during the stretching process. The hydrogel's brightness also dimmed during the stretching process. This is because the cross-sectional area of the PEC composite conductive hydrogel decreases as the stretch length increases. In Figure 7g, to further test the stability of the conductive properties of the hydrogel, the hydrogel was subjected to 1000 cycles of stretching at a strain of 70%, and the inset shows that the hydrogel has a relatively constant rate of change in resistance. This is due to the water evaporation from the hydrogel network during the test, increasing resistance. As shown in the inset of Figure 7g, the hydrogel maintained a stable resistivity during the pre-stretching period and after a long stretching period, further demonstrating the evaporation of water from the hydrogel during the stretching process.

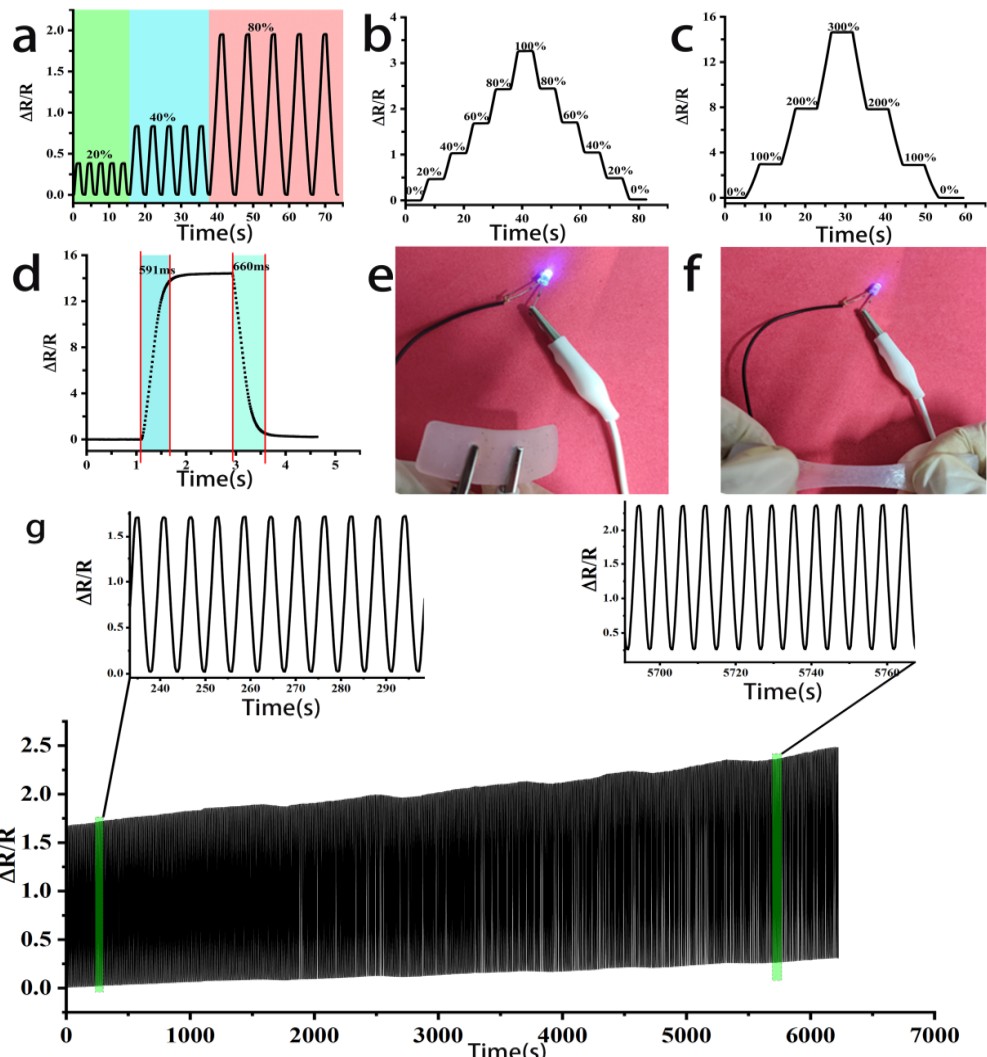

**Figure 7.** $PEC_{2.25\%}$ composite conductive hydrogel: (**a**) Resistivity change under 5 tensile cycles with strains of 20%, 40%, and 80%, respectively. (**b**) Stretch recovery at small gradients. (**c**) Stretch recovery at large gradients. (**d**) The reaction time of hydrogel. (**e**) The small bulb glows in normal conditions. (**f**) The small bulb is in stretch condition. (**g**) Change in resistivity of the hydrogel at 1000 stretch cycles.

*3.6. Applications in PEC Composite Conductive Hydrogel Strain Sensors*

To further study the electrical properties of the PEC composite conductive hydrogel, this paper fixed the hydrogel on the human finger joints, wrist, and elbow, as shown in Figure 8a. When the finger fixed with the PEC composite conductive hydrogel was bent to 30°, 60°, and 90°, the change in resistivity differed for different degrees of bending; the greater the bending degree, the greater the rate of change. The change in resistivity was also relatively stable for five consecutive bends at different bending levels. In addition, as shown in Figure 8b,c, the PEC composite conductive hydrogel was placed at the wrist and elbow, perpendicular to the body, and then bent to test; bending at different joints will lead to different degrees of change in the resistivity of the PEC composite conductive hydrogel. Therefore, different electrical signals can be identified to identify different joint flexion movements. As shown in Figure 8d–f, the PEC composite conductive hydrogel was placed on different parts of the dinosaur toy and the same part, undergoing different movements to identify different parts and different positions of the same part. The PEC composite conductive hydrogel was placed on the mouth of the dinosaur toy and tested by opening and closing the mouth, as well as on the side of the dinosaur tail and on top

of it by pulling and swinging it left and right, respectively, producing different electrical signals and different resistivity changes. Therefore, the change in resistivity of the PEC composite conductive hydrogel can be used to identify different parts and locations on the same item. The stretching and compressing properties of the PEC composite conductive hydrogel achieve this change in resistivity.

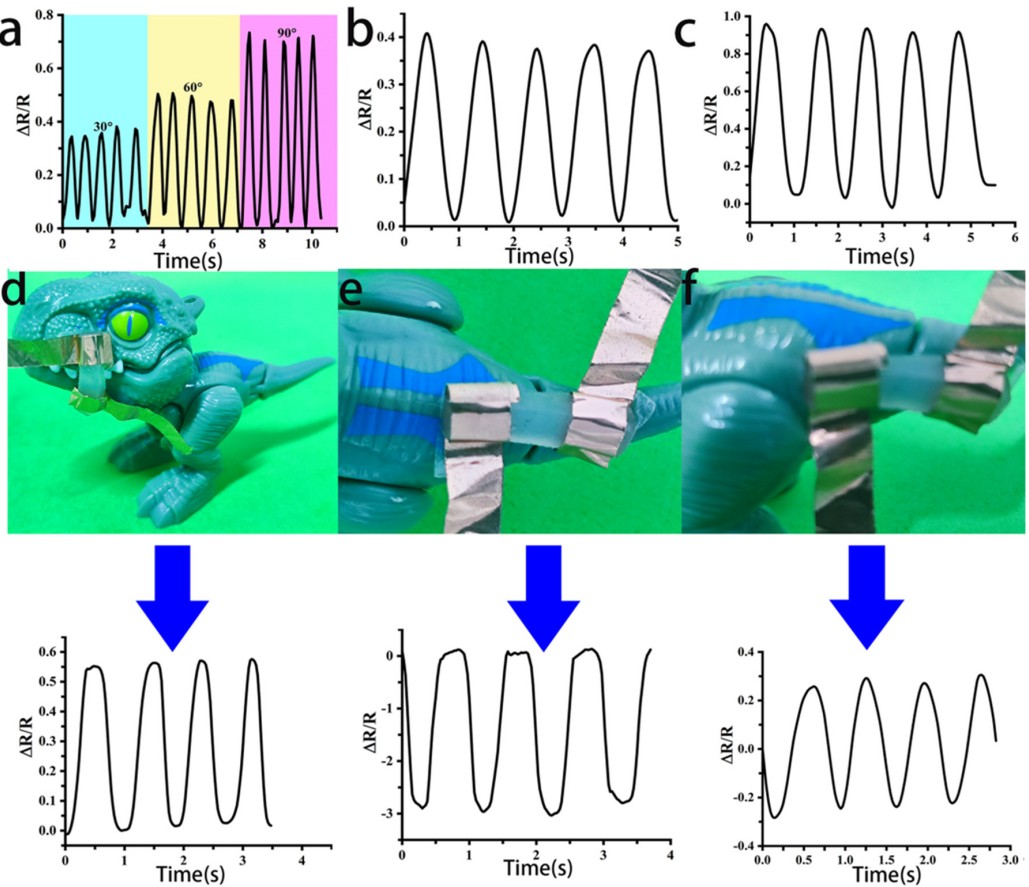

**Figure 8.** PEC$_{2.5\%}$ composite conductive hydrogel: (**a**) Change in resistivity at various degrees of finger flexion; (**b**) Change in resistivity of the hydrogel at the wrist; and (**c**) Change in resistivity of the hydrogel at the elbow. Hydrogel placed at the mouth of the dinosaur toy: (**d**) (open mouth, closed mouth); (**e**) Top of the tail (left and right swing); and (**f**) Change in resistivity at the side of the tail (left and right pull and pressure).

## 4. Conclusions

In this paper, the FTIR spectra showed that GX was coordinated with Al$^{3+}$ and then thermally polymerized with AAm in an aqueous solution, making the two intertwine. Adding CNWs gives the sensor high mechanical strength, excellent self-recovery performance, and a fast response time. The PEC composite conductive hydrogel has excellent mechanical properties due to many hydrogen bonds, coordination bonds, and cross-linking between the PAAM and the CNWs within the hydrogel. The maximum stress was 0.14 MPa, and the elongation at break was 707.1%, while the tensile stress of PAA/CNC hydrogel prepared by microwave incorporation according to Cheng et al., was 0.063 MPa and the elongation at break was 556% [32]. In this paper, the mechanical properties of the hydrogel prepared were improved. Adding NaCl to the hydrogel improves the coordination between Al$^{3+}$ and GX and enhances the electrical conductivity of the PEC composite conductive hydrogel, allowing it to have a fast reaction time (591 ms). The high-water absorption property of GX improves the hydrogel's swelling rate. The double helix structure of GX and CNWs makes the self-recovery properties of the fibers much higher. The hydrogel has



good fatigue resistance and restores the original sensing performance, and the self-recovery performance improves the PEC composite conductive hydrogel's lifetime. In this paper, the PEC composite conductive hydrogel was also used to detect the motion of human joints and a plastic toy. The raw materials used to prepare the PEC composite conductive hydrogel are non-toxic and non-hazardous. Moreover, the preparation is simple and easy to operate using the one-pot method. The hydrogels developed in this paper offer some novel concepts for high-tensile strength hydrogels.

**Author Contributions:** Conceptualization, Z.D. and X.L.; methodology, Z.D. and X.L.; validation, Z.D. and Y.W.; formal analysis, Z.D.; investigation, Y.W.; resources, X.L.; data curation, Z.D.; writing—original draft preparation, Z.D.; writing—review and editing, Z.D., X.L. and Y.W.; visualization, Y.W.; supervision, X.L. All authors have read and agreed to the published version of the manuscript.

**Funding:** This research received no external funding.

**Institutional Review Board Statement:** Not applicable.

**Informed Consent Statement:** Not applicable.

**Data Availability Statement:** The data presentend in this study are available on request from the corresponding author.

**Conflicts of Interest:** The authors declare no conflict of interest.

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
