# Peer review of "Preparation of Nanocellulose Whisker/Polyacrylamide/Xanthan Gum Double Network Conductive Hydrogels"

_coatings, doi:10.3390/coatings13050843_

Round 1

Reviewer 1 Report

Zhiwei Du describes the preparation of a composite interconnected hydrogel for applications as strain sensor. The author proposes an interesting one-pot preparation characterizing different aspects of the hydrogel, although the manuscript need a very deep improvement of English which in some parts make the paper very hard to understand. Moreover, some important aspects should be improved for its publication.

- Introduction and Abstract require a profound improvement of the definition of the aim of the work:

What is the optimum in terms of mechanical properties for a hydrogel applied in strain sensors?

What modification is expected with the introduction of cellulose?

What are the benefits of having a double network?

-       -   Line 72 -78 In the introduction there is no mention of what the author means by “first and second network”

-      -    Line 85 What is the dimension of the Cellulose nanowhiskers? What does 5 wt% referres to? Are there other components inside the Cellulose nanowhiskers?

-      -    Table 1 – The author should express the conc. of nanocellulose in % (wt) and name the composite accordingly (i.e PEC10%). Otherwise, it is difficult to evaluate what is the effect of the filler. Also the naming is phr (part per hundred rubber) is ok considering the total amount of polymer as 100%.

-   -       Figure 1: a and b) the figure is misleading since it seems that the AlCl3 solution is deposited on the mold. My suggestion is at least to change the color of the deposited liquid. c) it is a very good idea to make a scheme of the structure of the interconnected polymer, although the scheme is not clear. Visually it is not possible to detect any difference between the pre and post polymerization. The author should find a way to represent more clearly the structure and eventually make the chemical bonds clearer.

-      -    Lines 153 – 162 The structure detected at the SEM is the typical structure obtained by freeze-drying of polymers. It is not possible to assume that the pores are the same in the hydrogel since the structure was highly impacted by the freeze-drying process. Please rephrase the concept highlighting the fact that this is not the hydrogel structure, but a structure which is defined by the accretion of water crystals during freezing. In case, reference this work (Section 3.5 page 12) where this concept is described. Ladiè, Biomolecules 11.3 (2021): 389.  https://doi.org/10.3390/biom11030389

-    -      Lines 163 -166 EDS analysis cannot be used as a proof of complexation rather as evidence of distribution in defined area. Please clarify this concept and detail the area in which the Al is homogeneously distributed (200 um x 300 um?).

-       -   Lines 188 Define what the author means by small stress bonding.

-       -   Caption Figure 5d Please clarify what is the “reaction time”.

-       -   Caption Figure 5 Which is the sample measured? There is indication only for Figure 5a.

-       -   Lines 281- 283 the paper provides no proves of effective coordination. The author must include a chemical analysis of the structure to assure the crosslinking or, at least, reference some work that proved the same chemical structure.

Other minor corrections.

-         - Line 42 – 43 define acronyms

-        -  Figure 6 a, b ,c are missing.

-      -    256 repetition

 Please reply with a point-by-point letter.

Reviewer 2 Report

This paper concerns a preparation method for double-network conductive hydrogels. The authors used a conventional free radical polymerization approach to create a double-network hydrogel with xanthan gum carboxy-Al3+ as the second cross-linked network and polyacrylamide as the backbone network.

In my opinion, the idea of this manuscript is good; however, the experiments realized through this research are not enough to confirm such results.

Further, this paper contains some limitations that are not discussed or even mentioned and should be clarified, especially with regard to sample sizes and statistical considerations.

Reviewer 3 Report

Thanks to the Editor for inviting me to review the manuscript entitled “Preparation of nanocellulose whisker/polyacrylamide/xanthan gum double network conductive hydrogels.”

The authors reported the preparation of promising double network hydrogels based on cellulose nanowhiskers, polyacrylamide, and xanthan gum. The results have high reference values for related research and application. Therefore, I recommend accepting it after major revision.

Comments:

1-        Language should be thoroughly revised as some of the sentences are confusing and some errors can be found.

2-  The abstract should be modified and improved. In addition, some numerical results should be added to well reflect the content

3-        The importance and the applicability of the present study should be highlighted in the abstract and the introduction part.

4-        The originality and the novelty of the current study compared to previous recent works found in the open literature should be well clarified in the introduction part. The choice of nanocellulose whisker, polyacrylamide, and xanthan gum should be also addressed.

5-        The concepts of cellulose nanowhiskers and double network hydrogels should be well defined in the introduction part. In addition, I don’t agree with the authors about nanocellulose types (lines 49-50 page 2).

6-        Figure 1 should be replaced by a high-quality schematic presentation describing the preparation procedure of PEC hydrogels and their three-dimensional network structure will be interesting to improve the impact of the paper.

7-        Please correct the title of table 1.

8-        It will be helpful if the authors could provide the thermal characteristics of the investigated samples using TGA.

9-        In sum, the discussion of the results needs a deep improvement in order to improve the impact of the manuscript.

10-  Can the authors compare the reported results with other previous studies? It might also increase the impact of the results in this paper.

11-    The references part should be updated and some recent works in the field should be cited.

Round 2

Reviewer 1 Report

The author provided many improvements in the manuscript as requested and replied point-by-point. In this second version, the abstract was added of many details that are not useful in this part of the paper.

My suggestion is to improve the abstract only focusing on the idea, the strategy of experimental work and the impact on science.

Author Response

1.My suggestion is to improve the abstract only focusing on the idea, the strategy of experimental work and the impact on science.

Response 1: Thanks for your advice. Polish the paper. In the summary section, Firstly, 0.2g NaCl and 0.08g xanthan gum were added to the aqueous solution, then 2 ml AlCl3 (1wt%) was added under magnetic agitation, and then 8 g acrylamide, 4 mg N,N '-methylene diacrylamide,  80 mg sodium persulfate and different contents of cellulose nanocrystalline whiskers were added to form a hydrogel in  the oven at 50℃ for 2 h. "and the characterization test of hydrogels was added to lines 19-21 of the revision.

Reviewer 2 Report

This paper still contains some limitations that are not discussed or even mentioned and should be clarified, especially with regard to sample sizes and statistical considerations.

Author Response

This paper still contains some limitations that are not discussed or even mentioned and should be clarified, especially with regard to sample sizes and statistical considerations.

Response 1: Thanks for your advice. Polish the paper. In Section 3.4 of the revised draft, lines 287-290, limitations of hydrogel sample size are explained.

Reviewer 3 Report

The authors corrected all the necessary issues. I believe that the manuscript can be published in its current form.

Author Response

The authors corrected all the necessary issues. I believe that the manuscript can be published in its current form.

Response: Thank you for recognizing your work and efforts in this research, it is our honor to be recognized by you, and we will continue to work hard.